# Polyzwitterion–SiO_2_ Double-Network Polymer Electrolyte with High Strength and High Ionic Conductivity

**DOI:** 10.3390/polym15020466

**Published:** 2023-01-16

**Authors:** Lei Zhang, Haiqi Gao, Lixiang Guan, Yuchao Li, Qian Wang

**Affiliations:** 1School of Materials and Chemical Engineering, Chuzhou University, 1528 Fengle Avenue, Chuzhou 239099, China; 2State Key Laboratory of Chemistry and Utilization of Carbon Based Energy Resources, College of Chemistry, Xinjiang University, Urumqi 830017, China; 3Institute of Energy Innovation, College of Materials Science and Engineering, Taiyuan University of Technology, Taiyuan 030024, China; 4School of Materials Science and Engineering, Liaocheng University, Liaocheng 252000, China

**Keywords:** inorganic–organic double-network, polyzwitterion, polymer electrolyte, ionic conductivity

## Abstract

The key to developing high-performance polymer electrolytes (PEs) is to achieve their high strength and high ionic conductivity, but this is still challenging. Herein, we designed a new double-network PE based on the nonhydrolytic sol–gel reaction of tetraethyl orthosilicate and in situ polymerization of zwitterions. The as-prepared PE possesses high strength (0.75 Mpa) and high stretchability (560%) due to the efficient dissipation energy of the inorganic network and elastic characteristics of the polymer network. In addition, the highest ionic conductivity of the PE reaches 0.44 mS cm^−1^ at 30 °C owning to the construction of dynamic ion channels between the polyzwitterion segments and between the polyzwitterion segments and ionic liquids. Furthermore, the inorganic network can act as Lewis acid to adsorb trace impurities, resulting in a prepared electrolyte with a high electrochemical window over 5 V. The excellent interface compatibility of the as-prepared PE with a Li metal electrode is also confirmed. Our work provides new insights into the design and preparation of high-performance polymer-based electrolytes for solid-state energy storage devices.

## 1. Introduction

Polymer electrolytes (PEs) have attracted much attention recently due to their non-volatility, flame resistance, ease of processing, and low cost [1,2,3,4,5,6]. More importantly, PEs have the ability to inhibit the dendrite growth of solid metal (e.g., Na, Li) batteries, which are expected to fundamentally solve the safety problems of metal-based cells [7,8,9,10,11]. Whether it is a solid-state lithium metal battery or a solid-state sodium ion battery [12,13], polyethylene oxide (PEO) and its derivatives are currently the dominant PE matrices due to PEO’s good solubility for metal salts and its ability to transport metal ions [14,15,16]. However, PEO is known to be highly crystalline at room temperature and usually has a limited ability to transport metal ions; for example, the room temperature ionic conductivity of PEO–lithium salt systems is typically in the range of 10^−6^–10^−8^ S cm^−1^ [17], which is far from the desired ionic conductivity of 10^−3^–10^−4^ S cm^−1^ expected for PEs. In order to improve the room temperature ionic conductivity of PEO-based electrolytes, the key is to inhibit the crystallization of the polymer and thus improve the motility of the polymer chain segments. Physical blending [18,19,20], grafting [21], copolymerization [22,23], cross-linking [24], and branching [12] have been used to inhibit the crystallization of polymers. Although these methods can improve the ionic conductivity of PEs, the extent of their improvement is usually very limited, e.g., Appetecchi et al. made the room temperature ionic conductivity of the corresponding PEs reach ~1.1 × 10^−6^ S cm^−1^ using the method of blending modification [25], Niitani et al. made the room temperature ionic conductivity of PEs increase to ~1 × 10^−5^ S cm^−1^ using the strategy of copolymerization [26], and a PE with room temperature ionic conductivity of 8 × 10^−5^ S cm^−1^ was obtained using the branching method by wang et al [27]. To further improve the ionic conductivity of PEs, attempts have been made to introduce plasticizers such as liquid electrolytes and ionic liquids into PEs [28,29], which has significantly improved the room temperature ionic conductivity of PEs, but usually at the expense of the mechanical properties such as strength and tensile properties.

Materials with a double-network structure have been pioneered in the study of hydrogels [30], and better mechanical properties are usually obtained for polymers with a double-network structure compared to those with a single-network structure. The first network of materials with a dual-network structure usually consists of a rigid network in which there are hydrogen bonds, ionic bonds, or physical cross-linking points as sacrificial bonds, etc. The sacrificial bonds are reversibly broken and generated during the stretching and rebounding process as a mechanism to dissipate energy and improve the mechanical strength of the material [31]; the second network is usually a flexible polymer network connected by covalent bonds, which provides elasticity and maintains the basic architecture of the dual network [32]. In the field of ionogels, Kamio et al. first demonstrated that an ionogel based on an organic–inorganic double network has much higher mechanical strength than the single-network ionogel [33]. Inspired by this, Yu et al. constructed an organic–inorganic double-network solvate ionogel with high toughness (80 MPa) and high ionic conductivity (0.12 mS cm^−1^) for lithium-metal batteries [34]. However, there are few reports on organic–inorganic dual-network ionogel electrolytes, and the polymer network is limited to polyacrylamide, resulting in unsatisfactory mechanical properties, such as the strain of only ~170% that was reported by yu et al [34]. There is an urgent need to develop more types of new dual-network ionic gels.

Herein, we designed a new double-network polymer electrolyte (PE) based on the nonhydrolytic sol–gel reaction of tetraethyl orthosilicate and the in situ polymerization of zwitterions to synergistically achieve high strength, high tensile, and high ionic conductivity of PEs. Among them, the inorganic network is a physical crosslinking point composed of Si nanoparticles that can reversibly dissipate the energy generated by stretching, thus enhancing the tensile strength of the electrolyte; the elastic polyzwitterion network gives the electrolyte high tensile capacity. The ion–dipole forces between the polyzwitterion segments and between the polyzwitterion segments and ionic liquid in the organic network provide dynamic nano-conducting channels to facilitate ion transport, and the inorganic network can further enhance the electrochemical stability of the electrolyte, ultimately enabling the construction of high-performance electrolytes. Excellent interfacial compatibility of the PEs with lithium metal electrodes has also been demonstrated.

## 2. Experimental Section

### 2.1. Materials

2-methacryloyloxyethyl phosphorylcholine (Macklin, Shanghai, China), Tetraethyl orthosilicate (Aladdin, Shanghai, China), formic acid (FA, 88%, Aladdin) 1-ethyl-3-methylimidazoliumbis(trifluoromethylsulfonyl)imide (Aladdin, Shanghai, China), bis(trifluoro methane) sulfonimide lithium (Aladdin, Shanghai, China), 3-[Dimethyl-[2-(2-methylprop-2-enoyloxy)ethyl]azaniumyl]propane-1-sulfonate (Macklin, Shanghai, China), and 1-hydroxycyclohexyl phenyl ketone (Aladdin, Shanghai, China) were used directly without other treatment.

### 2.2. Synthesis of Organic–Inorganic Double-Network PEs

The double-network PEs were prepared using the sol–gel reaction of tetraethyl orthosilicate and in situ polymerization of zwitterions. Detailed information on the sample can be found in Appendix A. Typically, 2-methacryloyloxyethyl phosphorylcholine (MPC, 0.572 g, 1.94 mmol) and 3-[Dimethyl-[2-(2-methylprop-2-enoyloxy)ethyl]azaniumyl]propane-1-sulfonate (DPS, 0.064 g, 0.23 mmol) were dissolved in 1-ethyl-3-methylimidazoliumbis(trifluoromethylsulfonyl)imide (IL)/lithium salt (0.8 g, 30 wt% LiTFSI), followed by adding TEOS (120 μL), FA (160 μL), and a photoinitiator ( 1-hydroxycyclohexyl phenyl ketone, 0.018 mg, 0.09 mmol). The mixture was put into a homemade mold and underwent photopolymerization for 30 min, and then it was transferred to a 50 °C oven for 48 h. Finally, the double-network PE was obtained.

### 2.3. Characterization and Testing

Structural information on the monomers and PEs was recorded using FTIR spectroscopy (L1600400 Spectrum TWO DTGS, MA, USA). Thermal properties of the PEs were acquired using a thermal gravimetric analyzer (NETZSCH STA 2500, Selb, Germany) and differential scanning calorimetry (DSC, Shimadzu DSC-60A equipment, Tokyo, Japan). A Field Emission Scanning Electron Microscope (FESEM, Hitachi SU8010, Tokyo, Japan) was used to observe the microstructures of the PEs. Stress–strain curves of the PEs were obtained using the Instron 3300 electronic universal material testing system. The ionic conductivity (σ) of the PEs was obtained from impedance measurements using Zennium Electrochemical workstation (ZahnerEnnium) and calculated using σ = *l*/(*RS*), where *l* is the thickness of an electrolyte, *R* is bulk resistance, and *S* is the contact area between the stainless steel (SS) electrode and PE [35,36,37]. Linear sweep voltammetry was used to test the electrochemical stability of the PEs.

## 3. Results and Discussion

Figure 1a shows the preparation route of the novel organic–inorganic double-network polymer electrolyte based on the nonhydrolytic sol–gel reaction of tetraethyl orthosilicate and in situ polymerization of zwitterion (see the detailed preparation method in the experimental section). Figure 1b and 1c shows the IR spectra of Tetraethyl orthosilicate (TEOS), DPS, MPC, double-network PE (DPE) and single-network PE (SPE) form different wavelength ranges. According to the FTIR spectra, the obvious characteristic peaks of −CH=CH_2_ from MPC (3029 cm^−1^ (Figure 1b) and 1635 cm^−1^ (Figure 1c)) and DPS (3038 cm^−1^ (Figure 1b) and 1635 cm^−1^ (Figure 1c)) completely disappeared in the DPE (M9D1T30)) and SPE (M9D1T0), without SiO_2_ inorganic network), demonstrating the sufficient polymerization of zwitterions.

The thermal stability of related PEs was further acquired using a thermal gravimetric analyzer. Here, we denote the prepared PE by MxDyTz, where x, y, and z represent the weight ratios of MPC, DPS, and TEOS, respectively. Specifically, in this work, we prepared the samples M5D5T20, M5D0T20, M9D1T15, M9D1T20, M9D1T25, M9D1T30, and M9D1T35 to explore their differences in mechanical and electrochemical properties (see Appendix A). The TGA curves in Figure 2a indicate that the prepared PEs have a high thermal decomposition temperature of over 200 °C. When the content of inorganic components reaches a certain level, their ability to improve thermal stability is demonstrated. Especially, M9D1T30 shows a thermal decomposition temperature of ~300 °C, which is the highest among all the samples. The high thermal decomposition temperature of PEs indicates that they can be used over a wide range of temperatures.

The mechanical properties of electrolyte membranes, such as mechanical strength and stretchability, are among the most important indicators of safe, high-performance solid-state batteries, and we further investigated the effect of the content of each component on the mechanical properties. We first investigated the effect of amphoteric monomer content on the mechanical properties of PEs. Comparing the stress–strain curves of M10D0T20 (stress: 0.45 MPa; strain: 367%), M9D1T20 (stress: 0.55 MPa: strain: 439%) and M5D5T20 (stress: 0.30 MPa: strain: 195%), it can be seen that the stress and strain of the materials increase and then decrease with the increase of DPS content. At a fixed MPC and DPS content, i.e., the samples M9D1Tz (z = 15, 20, 25, 30 and 35), the strain of the material increases as the TEOS content increases, and overall, the strength of the material increases, and when the TEOS content exceeds 30 (i.e., corresponding to M9D1T35), both the strain and stress of the material decrease. The best sample (M9D1T30) can reach a strain of 569% and a stress of 0.75 MPa. It can be seen that the content of amphoteric monomers has an important influence on the mechanical properties, which is due to the large difference in the stiffness and flexibility of the corresponding two polymer segments, and only in a reasonable ratio can the common advantages of both be exploited. In addition, there is also a threshold value for the content of inorganic networks, which is due to the fact that too high a content of inorganic networks usually agglomerates and causes a decrease in the mechanical properties of the material.

The digital images of M9D1T20 before and after stretching are shown in Figure 2c. It is clearly seen that the film is highly stretchable. In addition, the SEM image in Figure 2d shows a uniform distribution of nanoparticles on the surface of the films, which can be attributed to the inorganic networks of SiO_2_ formed through the non-hydrolytic-sol-gel of TEOS.

Based on the mechanical property analysis, M9D1T30, M9D1T20, M10D0T20, and M5D5T20 were preferably selected to further test their ionic conductivity between 30 and 80 °C. Overall, the ionic conductivity of the PEs decreases with increasing DPS content, as can be seen from the curves of ionic conductivity and temperature (Figure 3a). For instance, the ionic conductivity of M9D1T30, M9D1T20, and M10D0T20 at 30 °C is 0.30, 0.31, and 0.44 mS cm^−1^, respectively. Specifically, M5D5T20 with the highest content of DPS shows the lowest conductivity in the whole temperature range (0.14 mS cm^−1^ at 30 °C). This can most likely be attributed to the fact that polyMPC and lithium ions have a more reasonable coordination interaction, which facilitates the migration of lithium ions. According to the fitted results, M9D1T30, M9D1T20, M10D0T20, and M5D5T20 show activation energies of 8.00, 6.68, 5.65, and 3.57 kJ mol^−1^, respectively, and all the corresponding curves follow the VTF model [23], which suggests that the polyzwitterion segments have an important role in regulating ion transport. We also compared the ionic conductivity of M9D1T20 and M9D1T30 throughout the temperature interval (Figure 3a) and found that the overall effect of the SiO_2_ content on the ionic conductivity is not significant. However, we found that the content of SiO_2_ strongly affects the electrochemical stability of the PE. As shown in Figure 3b, M9D1 without SiO_2_ has a very unstable electrochemical window (below 3 V). In contrast, M9DT20 exhibits a high electrochemical window of close to 5 V. This is due to the ability of SiO_2_ to act as Lewis acid and adsorb trace impurities [38], which facilitates the electrochemical stability. Figure 3c shows that the electrochemical windows of the electrolytes after the addition of the SiO_2_ inorganic networks all exceeded 4 V, and in contrast to the relationship with conductivity, the increase in DPS content is overall favorable for the electrochemical stability. Meanwhile, M9D1T30 also shows a wide electrochemical window of 4.9 V (Appendix A). In conclusion, M9D1T30 is the best sample in terms of mechanical and electrochemical properties. The material has a strain of 569%, a tensile strength of 0.75 MPa, and a high ionic conductivity of 0.30 mS/cm at 30 °C, which is much higher than that reported for other PEs [5,34].

The interfacial stability of Li/Li symmetric cells is one of the key indicators to assess whether the electrolyte can be applied to lithium metal batteries. As shown in Figure 3d, it can be seen that the interfacial resistance of the Li/M9D1T30/Li cell gradually increases in the initial 1–7 days, which is due to the formation of SEI film on the surface of lithium metal. Then, the interface reaches stability after 9 days of shelving, corresponding to an interfacial resistance of about 100 Ω, indicating good interfacial compatibility between the electrolyte and lithium metal. As also shown in Appendix A, Li/ Li/M9D1T20/Li also has a stable interface after 7 days, further confirming the advantages of using the novel double-network PEs.

To further understand the ionic transport mechanism of the as-prepared PEs, we further obtained the DSC curves of M9D1T30, M9D1T20, and M10D0T20. Regardless of the sample, there is only one melt peak belonging to the ionic liquid at about 17 °C [39], and no crystalline peak was observed in the sample, indicating that the sample is amorphous, and this amorphous feature facilitates the transport of lithium ions. In addition, there are ion–dipole interactions between the polyzwitterion segments and between the polyzwitterion segments and ionic liquid, which facilitate the formation of dynamic ion transport channels in the electrolyte. In addition, the interface between the inorganic network and the organic network also facilitates the formation of an interfacial transport layer, which promotes the transport of lithium ions. Therefore, we propose the ion transport model for this dual-network electrolyte (Figure 4b).

## 4. Conclusions

We designed and prepared a novel organic–inorganic dual-network electrolyte using the non-hydrolytic sol–gel reaction and in situ photopolymerization. The effects of organic and inorganic components on the mechanical properties, ionic conductivity, and electrochemical stability of the electrolytes were systematically investigated by regulating the content of organic and inorganic networks. A polyzwitterion–SiO_2_ double-network PE with high mechanical strength (tensile strength: 0.75 MPa), good tensile property (strain: 569%), high conductivity (0.30 mS/cm at 30 °C), and wide electrochemical window (close to 5 V) was prepared. The electrolyte also had good interfacial compatibility with lithium metal and was expected to be used in solid-state metal batteries. The possible ion transport mechanism of the electrolyte was further given using the phase transition analysis of the polymer electrolyte and the unique molecular conformation of the polyzwitterion segments. The present work provides new ideas for the design and synthesis of novel organic–inorganic dual-network PEs for high-performance solid-state energy storage devices.

## Figures and Tables

**Figure 1 polymers-15-00466-f001:**
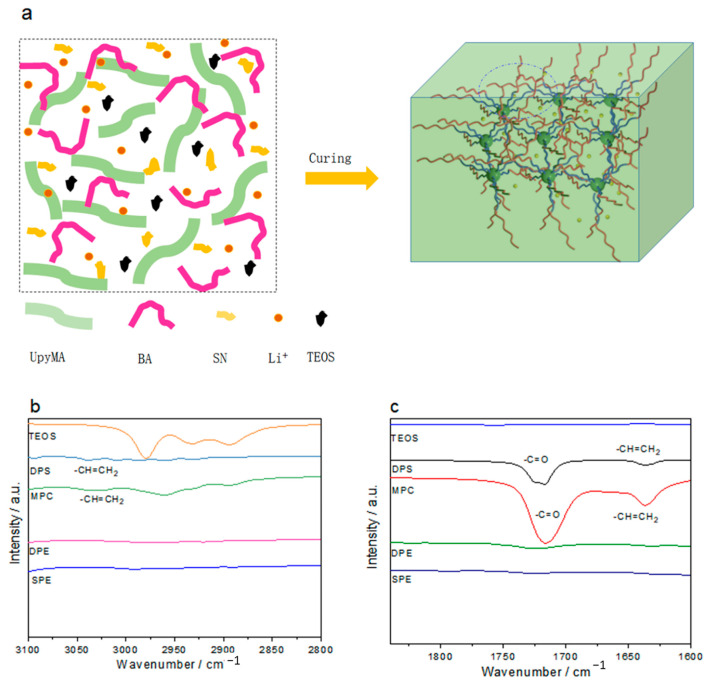
(**a**) Preparation route of a double-network PE. (**b**,**c**) FTIR characterization of related monomers and PEs.

**Figure 2 polymers-15-00466-f002:**
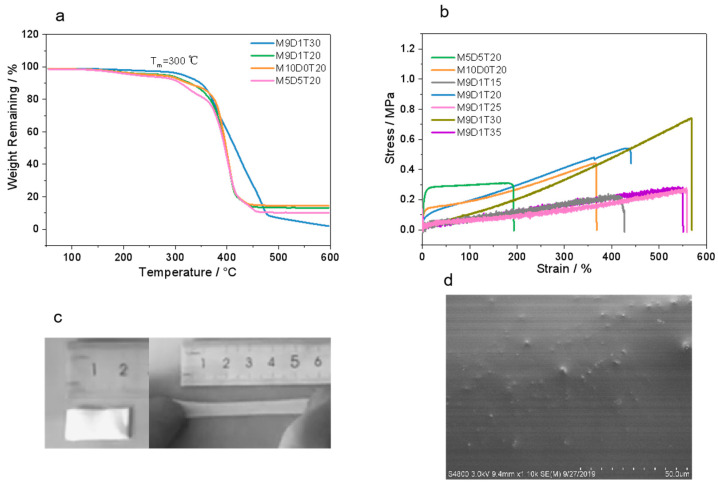
TGA curves (**a**) and Stress–strain curves (**b**) of double-network PEs with different ratios of TEOS and zwitterions. (**c**) Digital images of M9D1T20 before and after stretching. (**d**) Surface SEM image of M9D1T20.

**Figure 3 polymers-15-00466-f003:**
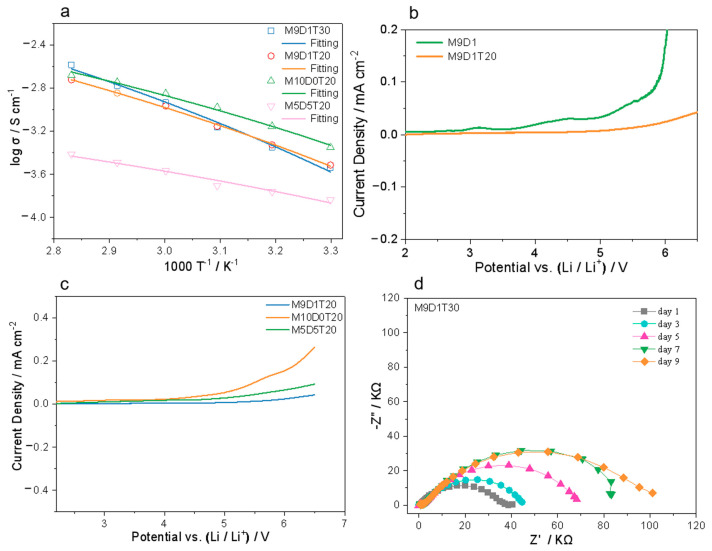
(**a**) Temperature dependencies of the ion conductivities of PEs. LSV curves (**b**,**c**) using double-network PEs with different ratios of TEOS and zwitterions. (**d**) EIS of Li/Li cell using M9D1T30.

**Figure 4 polymers-15-00466-f004:**
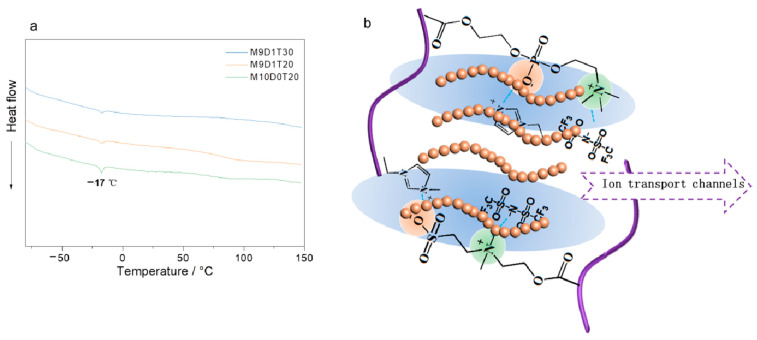
(**a**) DSC curves of double-network PEs with different ratios of TEOS and zwitterions. (**b**) Ion transport model of the prepared double-network PE.

## Data Availability

Not applicable.

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
