# Peer review of "Polyzwitterion–SiO2 Double-Network Polymer Electrolyte with High Strength and High Ionic Conductivity"

_polymers, 2023, doi:10.3390/polym15020466_

Round 1

Author Response

EditorThanadon - SupabunnapongPolymers 19 Dec., 2022 

Manuscript ID: polymers-2126142

Title: Polyzwitterion-SiO2 Double Network Polymer Electrolyte with High Strength and High Ionic Conductivity

Authors: Lei Zhang, Haiqi Gao, Yuchao Li* and Qian Wang*,

Dear Editor,

Re: Revised Manuscript for Polymers

Thank you very much for your kind efforts and your letter dated on 27 April 2022 regarding our manuscript (polymers-2126142).

We have considered the valuable remarks from the Reviewers seriously and revised our manuscript accordingly. Enclosed please kindly find our revised manuscript. We herein resubmit it for your kind consideration for publication in your prestigious journal—Polymers.

We also provide "Revised Manuscript with Highlighting Changes for Review Only" (.DOC), "Revised Manuscript" with no highlighting, and "Response Letter to the Comments" as requested for your kind consideration.

The responses to the comments are listed as following. The changes made in the revised manuscript are marked in red.

We believe now it is suitable for the present work to be published as an article in Polymers and will arouse substantial interests.

If there are any further queries, I can be reached by e-mail: qianwang19930825@163.com.

Thank you very much.

Yours sincerely,

Dr. Qian Wang

College of Materials Science and Engineering

Taiyuan University of Technology

Email: qianwang19930825@163.com; Mobile: +8618255053350

************************************************************

Point-by-point response to the reviewers’ comments

Reviewer: 1

Comment 1: “The double networks PEs were prepared via sol–gel reaction of tetraethyl orthosilicate and in-situ polymerization of zwitterions. Typically, MPC (0.572g, 1.94 mmol) and DPS (0.064 g, 0.23 mmol ) were dissolved in IL/lithium salt (0.8 g, 30 wt% LiTFSI), followed by adding TEOS (120 μL), FA (160 μL) and photoinitiator (0.018 mg, 0.09 mmol). The mixture was put into a homemade mold and photopolymerization for 30 min, and then transferred to a 50°C oven for 48 h. Finanlly, the double-network PE was obtained. How were the polymerization conditions adjusted, what was the photoinitiator?

Author Reply: We appreciate for the reviewer’s careful review. We adjusted the polymerization time from 10, 20, 30 to 40 min, it was found that 30 min can sufficient promote the polymerization of the active monomers. In addition, the photoinitiator is 1-hydroxycyclohexyl phenyl ketone, we have provided it in the experimental section.

Comment #2: The ionic conductivity (?) of PEs was obtained from impedance measurements using Zennium Electrochemical workstation (ZahnerEnnium) and calculated using ?=?/(?S), where l is the thickness of electrolytes, R is bulk resistance and S is the contact area between stainless steel (SS) electrode and PEs.[35-37] Linear sweep voltammetry was applied to test the electrochemical stability of PEs. why was a two-electrode system used?.”

Author Reply: Thank you very much for the good question. In the filed of PEs for LMBs, a two-electrode system is usually used to test the electrochemical stability of PEs due to the solid-state feature of PEs. In addition, the method of testing is now widely accepted. For instance, this approach was also adopted in a recently published work of ours (ACS Mater. Lett., 2022, 4, 1297).

Comment #3: The thermal stability of related PEs was further acquired via thermal gravimetric analyzer. Here, we denote the prepared PE by MxDyTz, where x, y and z represent the weight ratios of MPC, DPS and TEOS, respectively. Unfortunately, the authors do not translate all abbreviations used in the work. This is just one example.?

Author Reply: We appreciate for the reviewer’s careful review. “Specifically, in this work, we prepared samples M5D5T20, M5D0T20, M9D1T15, M9D1T20, M9D1T25, M9D1T30 and M9D1T35 to explore their different in mechanical and electrochemcial properties.” (Page 7). MxDyTz is used to represent the total samples in this work, but not just one example.  

Comment #4: Figure 1. (a) Preparation route of double networks PE. (a), (c) IR characterization of related monomers and PEs. What's the difference between IR and FTIR? Vague, no legend and not very clear. No analysis for Figures 2a and 2b. Poor quality drawings

Author Reply: We appreciate for the reviewer’s careful review. In this work, “IR” is the same as “FTIR”. We have revised “IR” to “FTIR” in the manuscript. We have further analyzed Figures 2a and 2b, and the quality of corresponding figures have been improved.

Comment #5: There is no table explaining the symbolism of the samples, so the interpretation of conductivity and impedance spectra is unclear.

Author Reply: We appreciate for the reviewer’s good advice. We have provide a table (Table S1) to explain the symbolish of the samples.

Table S1. Weight percentage of each component of MxDyTz, the strength, stress and  ionic conductivity also are shown.

MxDyTz

WMPC (%)

WDPS (%)

WTEOS (%)

Stress/ %

Strain/ Mpa

Ionic conductivity / mS cm-1 (30 oC)

M5D5T20

5

5

20

0.3

195

0.16

M10D0T20

10

0

20

0.45

367

0.44

M9D1T15

9

1

15

0.24

419

\

M9D1T20

9

1

20

0.55

439

0.31

M9D1T25

9

1

25

0.27

560

\

M9D1T30

9

1

30

0.75

569

0.3

M9D1T35

9

1

35

0.28

550

\

Comment #6: “Figure 4. (a) DSC curves of double networks PEs with different ratios of TEOS and zwitterions. (b) Ion transport model of the prepared double-network PE. Where does the shape of DSC curves come from - wrong graph, how does it relate to TG? bad description of the analysis.”

Author Reply: We appreciate for the reviewer’s careful review. The overall process of the DSC test is: the sample is heated from room temperature to 150 oC, then cooled to -90 oC, further heated to 150 oC, and then naturally cooled. In order to eliminate the thermal history, we use the curve from -90 oC to 150 oC as a reference, which is also the method used in numerous literatures (Chem. Eng. J., 2022, 440,, 135824; ACS Mater. Lett., 2022, 4, 1297).

        Therefore, according to the DSC curves, no crystalline peak was observed in the sample, indicating that the sample is amorphous, and this amorphous feature facilitates the transport of lithium ions.

        In addition, in fact, in Figure 4a, a small plateau can be seen around -60 oC, which can be attributed to the Tg peak of the materials.

Comment #7: The work requires many corrections, the authors did not justify the choice of techniques determining the success of the assumed goals. Unfortunately, the purpose itself is unclear and I don't know if the resulting PE would actually apply LIB. All work should be supported and then you can think about accepting it.

Author Reply: We appreciate the referee’s constructive comments. We have carefully revised the manuscript according the Reviewer’s suggestions. We analyzed the thermal, mechanical and electrochemcial properties of polyzwitterion-SiO2 double network polymer electrolytes with different weight ratio of polymers and SiO2 contents. The result indicates that MM9D1T30 is the best sample in terms of mechanical and electrochemical properties. The material has a strain of 569%, a tensile strength of 0.75 MPa, and a high ionic conductivity of 0.30 mS/cm at 30°C, which is a potential PE for LIBs.

        For PEs, the key that can be applied to LIBs is that the electrolyte has high electrochemical stability, high ionic conductivity and good film formation. Our optimized material is fully capable of meeting the above criteria, and therefore, the prepared PE can be applied to LIBs, and further results will be reported in the subsequent work.

Reviewer 2 Report

Overall, this is a well written manuscript, with clear motivation, well designed experimental evidence. Through rational design, the authors are able to achieve good mechanical performances and ionic conductivity. However, there are a few points that the authors could work on to improve the quality of this work before it gets accepted:

1. I would recommend that the authors make a table including all samples MxDyTz that appear in this work so that the experimental design is clearer.

2. Please also include a table showing modulus, strength, toughness, ionic conductivity, etc. 

3. The mechanical loading figure and conductivity figure (2-3) need to be improved by increasing line thickness. The current figures do not differentiate different samples very well due to the color contrast and line thickness. The authors need to do corresponding changes to make them better (Figure 1 is great).

Author Response

EditorThanadon - SupabunnapongPolymers 19 Dec., 2022 

Manuscript ID: polymers-2126142

Title: Polyzwitterion-SiO2 Double Network Polymer Electrolyte with High Strength and High Ionic Conductivity

Authors: Lei Zhang, Haiqi Gao, Yuchao Li* and Qian Wang*,

Dear Editor,

Re: Revised Manuscript for Polymers

Thank you very much for your kind efforts and your letter dated on 27 April 2022 regarding our manuscript (polymers-2126142).

We have considered the valuable remarks from the Reviewers seriously and revised our manuscript accordingly. Enclosed please kindly find our revised manuscript. We herein resubmit it for your kind consideration for publication in your prestigious journal—Polymers.

We also provide "Revised Manuscript with Highlighting Changes for Review Only" (.DOC), "Revised Manuscript" with no highlighting, and "Response Letter to the Comments" as requested for your kind consideration.

The responses to the comments are listed as following. The changes made in the revised manuscript are marked in red.

We believe now it is suitable for the present work to be published as an article in Polymers and will arouse substantial interests.

If there are any further queries, I can be reached by e-mail: qianwang19930825@163.com.

Thank you very much.

Yours sincerely,

Dr. Qian Wang

College of Materials Science and Engineering

Taiyuan University of Technology

Email: qianwang19930825@163.com; Mobile: +8618255053350

************************************************************

Reviewer 2:

Comment: “Overall, this is a well written manuscript, with clear motivation, well designed experimental evidence. Through rational design, the authors are able to achieve good mechanical performances and ionic conductivity. However, there are a few points that the authors could work on to improve the quality of this work before it gets accepted:

Author Reply: We appreciate very much for the insightful and constructive comments, which are helpful to improve the manuscript. We have considered very carefully the reviewer’s comments and revised the manuscript accordingly. These revisions have been marked in red in the revised manuscript. The answers for the comments are as follows.

Comment #1:I would recommend that the authors make a table including all samples MxDyTz that appear in this work so that the experimental design is clearer.”

Author Reply: Thanks for this very helpful suggestion. We have added Table S1 in the supporting information.

Table S1. Weight percentage of each component of MxDyTz, the strength, stress and  ionic conductivity also are shown.

MxDyTz

WMPC (%)

WDPS (%)

WTEOS (%)

Stress/ %

Strain/ Mpa

Ionic conductivity / mS cm-1 (30 oC)

M5D5T20

5

5

20

0.3

195

0.16

M10D0T20

10

0

20

0.45

367

0.44

M9D1T15

9

1

15

0.24

419

\

M9D1T20

9

1

20

0.55

439

0.31

M9D1T25

9

1

25

0.27

560

\

M9D1T30

9

1

30

0.75

569

0.3

M9D1T35

9

1

35

0.28

550

\

Comment #2: "Please also include a table showing modulus, strength, toughness, ionic conductivity, etc. ”

Author Reply: We appreciate the referee’s constructive comments. We have added it in Table S1.

Comment #3: "The mechanical loading figure and conductivity figure (2-3) need to be improved by increasing line thickness. The current figures do not differentiate different samples very well due to the color contrast and line thickness. The authors need to do corresponding changes to make them better (Figure 1 is great)..

Author Reply: Thanks very much for the very helpful comments. We have improved the Figures.

Round 2

Reviewer 1 Report

The work in my opinion has not been improved. The presentation of the average results and the description are still unreliable. The proof is the amount of corrections made to red (1%).

Author Response

Editor

Thanadon - Supabunnapong

Polymers 

02 January, 2023

Manuscript ID: polymers-2126142

Title: Polyzwitterion-SiO2 Double Network Polymer Electrolyte with High Strength and High Ionic Conductivity

Authors: Lei Zhang, Haiqi Gao, Yuchao Li* and Qian Wang*,

Dear Editor,

Re: Revised Manuscript for Polymers

Thank you very much for your kind efforts and your letter dated on 02 January 2023 regarding our manuscript (polymers-2126142).

We have considered the valuable remarks from the Reviewers seriously and revised our manuscript accordingly. Enclosed please kindly find our revised manuscript. We herein resubmit it for your kind consideration for publication in your prestigious journal—Polymers.

We also provide "Revised Manuscript with Highlighting Changes for Review Only" (.DOC), "Revised Manuscript" with no highlighting, and "Response Letter to the Comments" as requested for your kind consideration.

The responses to the comments are listed as following. The changes made in the revised manuscript are marked in red.

We believe now it is suitable for the present work to be published as an article in Polymers and will arouse substantial interests.

If there are any further queries, I can be reached by e-mail: qianwang19930825@163.com.

Thank you very much.

Yours sincerely,

Dr. Qian Wang

College of Materials Science and Engineering

Taiyuan University of Technology

Email: qianwang19930825@163.com; Mobile: +8618255053350

************************************************************

Point-by-point response to the reviewers’ comments

Reviewer: 1

Comment 1: The work in my opinion has not been improved. The presentation of the average results and the description are still unreliable. The proof is the amount of corrections made to red (1%).

Author Reply: We appreciate the referee’s constructive comments. All the questions have be carefully revised in our first revision. The manuscript have provided polymerization conditions, types of photoinitiators, explanation of why a two-electrode system was used to obtain the electrochemical stability of PE, definition of MxDyTz, more analysis of Figures 2a and 2b, details of MxDyTz, analysis of DSC curves, etc.

In fact, it is important to emphasize that in the present work we focus on how to achieve the synergistic regulation of mechanical properties and ionic conductivity of the material from the perspective of polymer material design. We propose a design strategy of organic-inorganic dual network, in which the brittle inorganic network dissipates energy and the elastic polymer network maintains the overall material integrity and deformation adaptability, thus achieving a synergistic regulation of the mechanical properties and ionic transport ability of polymeric materials. The effects of each component on mechanical properties and ionic conductivity are investigated in detail, and systematic mechanical, thermal and electrochemical properties are characterized to obtain the optimal ratio of performance, and the possible mechanisms of ion transport are further explored. The material is theoretically demonstrated to be a potential electrolyte material for use in the battery field. The practical application of the material as an electrolyte in batteries is not the concern of this work, and we will continue to report it in the subsequent work. Importantly, this work provides new ideas and guidance for the design of electrolyte materials at a later stage.

Round 3

Reviewer 1 Report

The work is average, but corrections have been made